# Preferences in ‘Jalapeño’ Pepper Attributes: A Choice Study in Mexico

**DOI:** 10.3390/foods10123111

**Published:** 2021-12-15

**Authors:** Blanca Isabel Sánchez-Toledano, Venancio Cuevas-Reyes, Zein Kallas, Jorge A. Zegbe

**Affiliations:** 1Instituto Nacional de Investigaciones Forestales, Agrícolas y Pecuarias, Campo Experimental Zacatecas, Calera de V.R., Fresnillo 98500, Zacatecas, Mexico; sanchez.blanca@inifap.gob.mx; 2Instituto Nacional de Investigaciones Forestales, Agrícolas y Pecuarias, Campo Experimental Valle de México, Texcoco 56250, Edo. de Mexico, Mexico; cuevas.venancio@inifap.gob.mx; 3Centre for Agro-Food Economy and Development (CREDA-UPC-IRTA) Parc Mediterrani de la Tecnologia, Edifici ESAB C/Esteve Terrades, Casteldefells, 08860 Barcelona, Spain

**Keywords:** *Capsicum annuum* ‘Jalapeño’, consumer preferences, willingness to pay, generalized multinomial logit model

## Abstract

Background: According to Mexican growers of ‘Jalapeño’ peppers, its commercialization is the primary limitation. Thus, consumer knowledge is critical to develop added-value strategies. The objective of this study was to identify ‘Jalapeño’ quality attributes to determine consumer preferences and willingness to pay, based on socioeconomic characteristics. Methods: A nationwide face-to-face survey was carried out using the discrete choice experiment method. The survey included 1200 consumers stratified by gender, age and region. Results: Heterogeneity analysis using the probabilistic segmentation model revealed three types of consumers: A price-sensitive segment, non-demanding consumers without specific preferences and selective consumers with a preference shifted toward specific ‘Jalapeño’ characteristics. Thus, detail-oriented producers must compete through price strategies, based on the marketplace (markets on wheels, grocery stores, or supermarkets) and through some quality attributes preferred by selective consumers. Therefore, results suggest that farmers should grow the correct varieties with appropriate agronomic management to cope consumer preferences. Conclusions: This paper contributes to the growing body of the ‘Jalapeño’ literature by explicitly investigating consumer preferences and willingness to pay for them.

## 1. Introduction

Chili (*Capsicum* spp.) is a commonly cultivated vegetable worldwide, with a production of 36,771,482 t [1]. Worldwide chili yield has increased from 15.5 t ha^−1^ in 2008 to 18.5 t ha^−1^ in 2018. This increase is consistent with the goals of world food security programs [2] because it is estimated that the food demand *per capita* should grow 4% for the next decade [3].

In Mexico, chili pepper cultivation has major social, economic and cultural importance. There are over 50,000 producers that employ ~15 million workers, making chili production a primary source of family income in rural areas [4]. Additionally, in this country, annual chili pepper production in Mexico was estimated at 3,200,000 t [4] and consisted of over 100 varieties distributed nationwide. Chili pepper varieties can be divided into two major groups: 22 varieties for fresh consumption and 12 for dry consumption. ‘Jalapeño’ peppers (*Capsicum annuum*) represent a third of Mexican pepper production (31%) and are sold fresh. In 2010, 33,000 ha were planted with ‘Jalapeño’, but this area decreased by 11.2% in 2020 [3]. The decrease has been attributed to various factors, including low benefit, over yield, excessive intermediaries in the supply chain, poor marketing and insufficient agricultural credits, among others [5]. In addition, new challenges have emerged as supply chains shift their focus to satisfy consumer demands directly. For instants, the epidemic caused by SARS-CoV-2 (COVID-19) has shifted all food supply chains in order to get fresher products from the field to doorsteps [6]. Supply and added-value strategies require more information on consumer behavior, in particular in these pandemic times [6]. In Mexico, chili peppers have been a basic ingredient in the Mexican diet since pre-Hispanic times, along with products derived from corn (*Zea maiz* L.), pumpkin (*Cucurbita pepo* L.) and beans (*Phaseolus vulgaris* L.) [7]. In agri-food chains, consumers are regarded as end users; thus, consumer behavior and characteristics are relevant to market-driven organizations that manage supply chains [8,9]. Moreover, consumer behaviors such as decision-making are influenced by internal and external factors, which can be rational or irrational. Therefore, consumer decisions affect the market and economic growth [10].

Consumers evaluate goods and services using three main criteria: (1) intrinsic, (2) extrinsic and (3) psychological attributes [11]. Intrinsic attributes such as taste, composition, color, smell, size, quantity, design, packaging and labeling are perceived directly. Extrinsic attributes are related to assortment, range, price and usability. Finally, psychological attributes include reputation, credence certifications, brand and perceived quality. Therefore, consumption is not driven entirely to benefits provided by a good, but also to a cost-sacrifice relation, making product alternatives the result of a subjective cost–benefit exchange.

Changes in consumer demand over the last decade have increased research on food quality [12]. Analysis of change in agri-food markets highlights product quality as an important parameter [13]. Therefore, commitment to quality has become a reliable growth opportunity in international markets [14]. Likewise, the meaning of ‘quality’ to particular groups of consumers has become a relevant factor in the purchasing process [15]. Thus, willingness to pay (WTP) for goods or services largely depends on their perceived quality, especially for food products [16].

In this context, it is essential for farmers and industry stakeholders to determine and understand the attributes that generate the highest quality level to allow efficient use of resources [17]. In Mexico, the published research that incorporates consumer perspectives and preferences in the agricultural sector is scarce and rarely developed [18,19,20,21]. Therefore, the objectives of this research were to identify sought-after ‘Jalapeño’ quality attributes and then evaluate willingness to pay (WTP) based on consumer socioeconomic characteristics. It is expected that this research will contribute to ‘Jalapeño’ breeding programs by including a social perspective in the development of agricultural and marketing strategies for promoting ‘Jalapeño’ consumption.

## 2. Materials and Methods

### 2.1. Data Collection and Sampling

Data were collected from November 2019 to March 2020 using a semi-structured survey with 21-question [22] grouped in blocks. Before data collection, pilot tests were conducted to ensure question clarity and avoid interview mistakes (*n* = 30). Although Mexico has a population of 130 million, this research only considered the adult segment (age 20 and older), equivalent to 67 million people [23]. Finite population sampling suggested a sample size of 1040. However, 1200 questionnaires were randomly administered to generate a sampling error of 4% and a confidence level of 99%. Data from the National Institute of Statistics and Geography (INEGI) was used for sampling calculations (Table 1). Selected individuals also became participants in the choice experiment used to define relevant factors for ‘Jalapeño’ consumption. The questionnaire applied was validated and approved by a social science ethical committee. It was conducted according to the principles given in the Declaration of Helsinki, with particular care to protect personal information as required by Mexican regulations. Before applying the survey, the participants over 20-year were contacted outside of market on wheels, markets and supermarkets received an explanation of the experiment and signed a consent form, which was read aloud. The questionnaire was read to the participants by the researchers and it took around 40 min each.

### 2.2. The Discrete Choice Experiment: Theory and Modelling Approach

Consumer preferences for ‘Jalapeño’ attributes were analyzed using the discrete choice experiment method. Choice experiments were originally used in communication and transport research [20,24,25]. Subsequently, they were adopted in other research areas such as environmental assessment [26,27], market research [28,29,30], plant and animal improvement programs [31,32,33], environmental and consumer studies [34,35] and, recently, in agricultural value chain research [36,37,38]. Choice experiments are based on Lancaster’s consumer behavior theory and McFadden’s random utility theory [39,40]. According to these authors, consumer utility derives from perceived product attributes, rather than from the product itself. Consequently, a product is defined as a set of attributes with certain characteristics and individual choice reflects a combination of attributes that maximizes subjective utility. In the contingent choice model (p. 51 [41]), subjects choose a good from a set of alternatives to mimic market conditions [41]. In this context, the indirect utility function for each set of alternatives consists of three components: (1) the product attributes Zij, (2) the socioeconomic characteristics Si and (3) the income Yi. Individual *i* will prefer alternative h, rather than alternative j, if it has superior utility over other available alternatives within choice set C; that is, if *Uih* > *Uij* > *∀h* ≠ *j*; *h*, *j* ∈ C.

Moreover, all alternatives ensure an utility function with two main components: A systematic component (observable) and a random error term (non-observable) [42]:*U_jn_* = *V_jn_*
*Z_ij_*, *S_i_* + ɛ*_jn_*(1)
where *U_jn_* is the *j*-th utility of alternative to *n*-th subject, *V_jn_* is the systematic component of the utility, *Z_ij_* is the *j*-th vector of attributes of alternative, *S_i_* is the *n*-th vector of socio-economic characteristics of the subject and ɛ is a random term that is inversely related to a scale term (σ_n_).

The multinomial logit model (MNL) was used to formalize the decision-making process of subjects in their selection of the most preferred alternative [40]. Among various modeling approaches that include the scale heterogeneity specification is the generalized multinomial logit model (GMNL) [43]. According to this model, an individual’s utility (n) for selecting an alternative (*j*) in a choice set (*t*), are given by:*U_njt_* = [σ*_n_*β + γ*n_n_* + (1−γ) σ*_n_n_n_*] + ϵ*_njt_*(2)
where γ is a mixing parameter between 0 and 1, whose value represents the degree of independence or interaction between the scale term σ*_n_* and the heterogeneity around the attributes’ estimates (*n_n_*). The term σ*_n_* follows a log-normal distribution with mean equal to 1 and standard deviation τ. The GMNL estimates the τ term, which captures scale heterogeneity across respondents. According to the GMNL model, the WTP is directly estimated in the mode. This estimation procedure reduces the probability of excessively large WTP values, produces better data fitting and allows the analyst major control over the WTP distribution [44].

### 2.3. Latent Class Analysis

Heterogeneity in consumption behavior among subjects was assessed using the latent class analysis (LCA) approach [45]. The latent class models assign participants to behavioral groups or latent classes, which explain differences and homogeneity [46]. The “best” number of classes to be extracted is based on a comparison of the Bayesian information criterion (BIC), *r*^2^ and outcome probability [47]. The LCA was applied here to identify different consumer segments and ‘Jalapeño’ attribute levels. More details on this statistical tool are available [48]. Subsequently, collected data was used to perform a one-factor ANOVA test. This allowed simultaneous study of differences at *p* ≤ 0.05. The information was analyzed with SPSS 21.0 software.

### 2.4. The Discrete Choice Experiment: Empirical Applications

Prior to the experimental design of the choice sets (C), a discussion similar to a focus groups session was held with ‘Jalapeño’ researcher experts from the Instituto Nacional de Investigaciones Forestales, Agrícolas y Pecuarias, ‘Jalapeño’ growers and consumer regional associations (*n* = 25). This discussion identified the most important ‘Jalapeño’ attributes to consumers. Three attributes were selected to build the experimental design (Table 2).

The ’Jalapeño’ price per kg was determined according to market prices as observed in several establishments, with an additional 20% variation on the extreme values. Three ‘Jalapeño’ sizes were chosen to represent those available in the market. Pungency degree was selected because it is a decisive attribute for repeated purchases. Capsaicin and dihydrocapsaicin cause 90% of pungency in ‘Jalapeño’ peppers [49,50] and therefore, three pungency levels were assessed (Table 2).

The total number of combinations of the attributes was 46656, as determined by LMA, where L is the number of levels (4), M is the available alternatives (3) and A is the number of attributes (3).

Combinations were eventually reduced through an optimal and efficient experimental design that reduced the estimated errors using the Ngene software [51]. Furthermore, choices available to participants were decreased through block division. To ensure that block distribution was random and uncorrelated to attributes, blocks were considered as an additional attribute during the experiment [52]. The final experiment consisted of 32 products, or combinations of alternatives, that were distributed in four blocks with eight cards each, an example of a choice set is shown in Table 3.

During the face-to-face interview, the discrete choice experiment procedures and contents were explained in writing and orally to all participants. A pilot survey was administered to verify understanding, which suggested that small groups facilitate explanation. In addition, the alternative “none of the above” was added to ensure compliance to the demand theory, in which a no-choice option is possible, allowing for more accurate results [53]. Incorporating the opt-out option was necessary, as consumers often delay consumption in anticipation of products that better fit their expectations (improved attributes: price, brand, presentation) or due to a lack of satisfaction [54]. Including, “no choice” as an option can improve prediction of the performance of new products in the market [55]. The information was analyzed with the statistical software extension package Nlogit.

## 3. Results and Discussion

### 3.1. Sample Description

The participants were varied in gender, age, education and income. In Mexico, women are responsible for 60% of grocery purchasing [56]. In this survey, purchasing was attributed mainly to women (66%). The largest age groups were adults between 41 and 60 years (43.5%) and adults between 18 and 29 years (25%). Thus, sample age and gender values were consistent with official population statistics [23] (Table 4). The sample had relatively more education than the population as a whole [23]. About 46.1% of the sample had a monthly income below $256.4 USD, which is consistent with the current average income *per capita* in Mexico.

### 3.2. Consumer Preferences on ‘Jalapeño’ Attributes

The GMNL model provided results in the WTP-space (Table 5). The model showed a goodness-of-fit with an acceptable value of McFadden pseudo *r*^2^ (0.24), similar to other studies that analyzed consumer’s preferences through choice experiments [57]. The log likelihood ratio was also highly significant at 99%. Results showed that the estimated coefficients of the majority of attribute levels were statistically significant. This confirms that most of attributes and levels considered in the model are significant and essential to predict consumer preferences.

The estimated parameters show a negative relationship between consumer utility and both ‘Jalapeño’ size and pungency. Thus, for every 1000 Scoville units’ pungency increase, on average, the market price reduces by 0.19 USD/kg. In contrast, culinary demand and economic importance of habanero peppers depends on their high degree of pungency [50]. Similarly, pepper consumers in Oaxaca, Mexico, demand peppers with higher concentrations of capsaicinoids [58].

Regarding fruit size, for each unit of increase in fruit size, on average, the price decreases by 0.14 USD/kg. Medium-sized ‘Jalapeños’ are preferred by consumers as they can be consumed at home quickly [59]. In addition, large families look for small-sized ‘Jalapeño’s for two reasons: There will be more peppers units per kilogram and they can be cold-stored to be used as required [60].

In this context, consumer utility decreases with the price increase. That is, at lower costs, the number of ‘Jalapeño’ purchases increase. The outputs of the market for ‘Jalapeño’ peppers agreed with the principles of the economic theory of demand. This behavior is explained by the frequent use of this fresh product in Mexican cuisine, unlike that used of other full-processed products such as cheese, where consumption is determined by a set of social, cultural and economic features [61]. ‘Jalapeño’ peppers’ moderate pungency and year-round availability also boosts its consumption in domestic and international markets. For instance, in the United States, a survey administered to 1104 consumers in 2012 found that ‘Jalapeño’ peppers were the most popular and preferred product among seven types of hot peppers [62].

### 3.3. Consumer Heterogeneity towards ‘Jalapeño’ Peppers

The outputs of the estimated latent class model revealed three consumer segments based on preferences (Table 6). Calculations were performed to determine the optimal number of segments using the BIC, pseudo *r*^2^ and probability for each segment [63]. The latent class model with three classes was selected as the best option. Based on the probability, 32% of participants were price-sensitive; while 51% were indifferent towards specific attributes and 15% had a very specific acceptance pattern.

Consumers from the first segment were mainly affected by price fluctuations; that is, consumers mostly considered income and expenses in the purchasing decision, as mentioned elsewhere [64]. Consumers in the second latent class had no specific preferences for ‘Jalapeño’ attributes and were less concerned about price. Lastly, consumers from the third latent class had a pronounced preference toward small size, lower pungency and average price. Finally, price was significant and negatively related to the three classes, highlighting consumer sensibility to ‘Jalapeño’ prices.

### 3.4. Profile of Consumer Segments

The three consumer latent classes identified were studied further to understand the behaviors that determine consumption for each consumer profile and to build new competitive market strategies. Such information allows stakeholder decisions along the added-value chain to efficiently address each type of preference (Table 7).

Price-sensitive consumers make the purchase decision at informal, temporarily established markets known as “markets on wheels”. Economic crisis motivates this consumer segment, who focus on price and value due to their low incomes. Food demand is predominantly price-driven, but assessing price sensitivity is increasingly driven by heterogeneous attributes [65,66]. Households in this group had from one to three consumers and considered ‘Jalapeño’ origin, size and pungency important. However, they were not willing to pay a premium for these attributes. Additionally, these consumers would purchase bell peppers, but would not consider processed ‘Jalapeño’ products as a substitute. This group had also a monthly income from $254 to $550 USD and a high school education level.

The non-demanding consumers viewed ‘Jalapeño’ attributes with indifference. In this group, consumers purchased ‘Jalapeños’ in markets on wheels, although supermarkets were also an option, as they derived utility from a quality-price relation rather than price itself. Male participants were college-educated with monthly incomes from $551 to $770 USD and averaged 30 years old. Better knowledge of what ‘Jalapeño’ consumers need and deem important and valuable is essential both to communicate salient features of existing product lines and to direct properly the selection and development of new lines to better meet customers’ needs. Better-informed customers make more informed and rational decisions, providing increased satisfaction for them and pushing the industry as a whole toward efficiency and qualitative improvement [67].

Selective consumers with specific attribute preferences purchased fresh or processed ‘Jalapeño’s at established supermarkets, where almost all products can be purchased at all times. These consumers avoid purchasing from informal and other kinds of establishments. This behavior is attributed to long working hours and poor work–life balance. While ‘Jalapeño’ quality attributes were ignored by these consumers, but at the same time, they weighed for the readily available products. Globalization has undermined healthier food options by putting small food-supply chains and local producers at risk. Therefore, technical solutions aimed at improving short food-supply chains and local production are urgent and potentially life-saving [68]. This segment is also interested in the product’s origin and in innovative ‘Jalapeño’ products; thus, local production can potentially benefit. Consumers with a local identity show lower price sensitivity [69]. Therefore, growers can increase their market share by adopting a local producer identity. This is not a novelty: in fact, respondents often consider a local producer identity as a realistic and reliable quality clue [70]. The development of a sustainable food system is accompanied by local sustainable development policies that take into account different aspects of sustainability [71].

Current ‘Jalapeño’ supply allows consumers to choose from many ‘Jalapeño’ varieties and options. Thus, detail-oriented producers must compete through price-based strategies [72] based on the marketplace (markets on wheels, grocery stores, or supermarkets). Furthermore, consumer behavior illustrates income level differences and a clear understanding of the advantages and disadvantages of each market type and its offerings [73]. Additionally, the pandemic SARS-CoV-2 (COVID19) may modify markets, prices and consumer preferences; therefore, further studies must be conducted to explore this hypothesis.

## 4. Conclusions

Although ‘Jalapeño’ is grown widely in Mexico and the ‘Jalapeño’ industry has been around for decades, growers have neglected consumer preferences regarding ‘Jalapeño’ attributes. As a result, connection between primary growers and end users has been disrupted. Our research demonstrates the importance of consumer preferences and behavior on ‘Jalapeño’ attributes. Consumers preferred moderately spicy (6000 USc) and medium-sized (6.25 cm) ‘Jalapeños’. Therefore, growers must use appropriate varieties and crop management techniques to achieve these results.

Analysis of preference heterogeneity among ‘Jalapeño’ consumers in Mexico revealed three consumer profiles with respect to price: Price sensitivity, non-demanding (indifferent) and selective. Customer classification by segments allows growers to focus efforts into less demanding segments or develop new market strategies. Moreover, new policies encouraging ‘Jalapeño’ cultivation must consider each segment’s characteristics and preferences. From a business perspective, these results suggest an area of opportunity, in which ‘Jalapeño’s’ growers may ask for new varieties and crop technologies to engage different market segments.

## 5. Patents

There are no patents resulting from the work reported in this manuscript.

## Figures and Tables

**Table 1 foods-10-03111-t001:** Survey data sheet.

Information Collected	Experimental Period (Nov. 19–Mar. 2020).
Population	‘Jalapeño’ consumers in Mexico.
Universe	67 million [23].
Confidence level	99 × 100
Possible margin error	±4 per 100
Sample	1200
Sampling type	Simple random

**Table 2 foods-10-03111-t002:** Attributes and levels from ‘Jalapeño’ fruit.

Attribute	Attribute Symbol	Level	Level Symbol
Price	A1	51 cents USD/kg	L_1.1_
		56 cents USD/kg	L_1.2_
		1.07 USD/kg	L_1.3_
		2.05 USD/kg	L_1.4_
Fruit size	A2	Medium (6.25 cm)	L_2.1_
		Large (9 cm)	L_2.2_
		Jumbo (10 cm)	L_2.3_
Pungency degree	A3	Moderately spicy (6000 USc)	L_3.1_
		Spicy (11,000 USc)	L_3.2_
		Very spicy (17,500 USc)	L_3.3_

**Table 3 foods-10-03111-t003:** Subjective and discrete choice scenarios regarding ‘Jalapeño’ attributes.

Card 1	Option A	Option B	Option C
Size	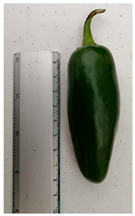 Large (9 cm)	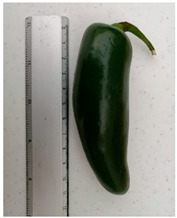 Jumbo (10 cm)	None of the above
Price	Less than 51 cents	More than 2.05 USD	
Pungency	Very spicy (17500 USc)	Spicy (11000 USc)	
I would choose			

**Table 4 foods-10-03111-t004:** Sociodemographic characteristics (%) of ‘Jalapeño’ consumers in Mexico.

Sampled Population Characteristics	Sample (*n* = 1200)	Total Population (Mexico)
Gender		
Female	66.0	51.4
Male	34.0	48.6
Age (years)		
18–29	25.0	25.6
30–40	20.4	14.4
41–60	43.5	21.8
>60	11.1	10.5
Education level		
Primary or lower	11.5	31.2
Secondary school	20.2	27.9
High school	26.8	21.7
University	37.7	18.6
Graduate	3.8	8
Income level in USD		
<251	46.1	29.0
251–550	31.7	32.0
551–770	13.1	34.0
771–1100	6.0	3.1
1101–1500	1.8	1.0
1501 and over	1.3	0.9

**Table 5 foods-10-03111-t005:** The generalized multinomial logit model in willingness to pay-space model for ‘Jalapeño’ consumers.

Attribute	$\hat{\beta }$	Probability Value
	Random parameters in utility functions	
Size	−0.03	0.015
Pungency	−0.04	0.00
	Non-random parameters in utility functions	
Price	−0.05	0.000
No	−3.85	0.000
	Scale parameters	
Variance parameter tau (τ +) in sacle parameter	0.15	0.000
Weighting parameter gamma (γ ++) in GMX model	0.82	0.000
NsSize	0.17	0.000
NsPungency	0.12	0.000
Log likelihood function	−5367.1	
Restricted log likelihood	−7031.2	
Pseudo-*r*^2^	0.24	

+ Tau estimate capture the scale heterogeneity across consumers; ++ The weighting parameter is a mixing parameter and its value determines the level of mixing or interaction between the scale heterogeneity and the parameter heterogeneity.

**Table 6 foods-10-03111-t006:** Latent class model of ‘Jalapeño’ consumers in Mexico.

Latent Class	Coefficient	Probability Value
Price sensitive (Latent Class 1)	Class 1, utility parameters	
Size	−0.02	0.46
Pungency	−0.01	0.36
Price	−0.20	0.00
NO	−8.42	0.00
Attribute-indifferent (Latent Class 2)	Class 2, utility parameters	
Size	−0.02	0.20
Pungency	−0.00	0.14
Price	−0.01	0.00
NO	−3.09	0.00
Attribute-specific preferences (Latent Class 3)	Class 3, utility parameters	
Size	−0.06	0.09
Pungency	−0.14	0.00
Price	−0.07	0.00
NO	−3.03	0.00
Estimated latent class probabilities		
Probability	0.32	
Probability	0.51	
Probability	0.15	
Log likelihood function	−5155.14	
Restricted log likelihood	−7031.11	
*r* ^2^	0.26	

**Table 7 foods-10-03111-t007:** Key parameters for differentiating consumer segments.

		Consumers	
Parameters	Price-Sensitive	Non-Demanding (Indifferent)	Selective
Purchase location	Market on wheels ^a,^*	Market and supermarket ^a,b^	Supermarket ^b^
Purchase quantity	0.5 kg or less ^b^	0.5 to 1 kg ^a,b^	1 kg ^a^
No. of relatives who consume ‘Jalapeño’	1 to 3 ^b^	1 to 3 ^b^	4 to 6 ^a^
‘Jalapeño’ source	Important ^b^	Indifferent ^c^	Very important ^a^
Customized preference	Probable ^b^	Indifferent ^c^	Very likely ^a^
Consideration for processed products	Probable ^b^	Indifferent ^c^	Very likely ^a^
Substitutes	Bell peppers ^a^	Bell and tree peppers ^b^	Tree peppers ^b^
Agro-industrial product of preference	Snack ^a^	Sauce ^b^	Cheese ^c^
Monthly income	251 to 550 USD ^c^	551 to 770 USD ^b^	771 to 1100 USD ^a^
Education	High school ^b^	University ^a^	University ^a^
Occupation	Housewives ^b^	Office worker ^a^	Office worker ^a^
Gender	Female ^a^	Male ^b^	Female ^a^
Age	52 ^a^	30 ^c^	38 ^b^
Consumer percentage of the sample	32	51	15

* For each parameter, consumer segments within rows followed by different letters are statistically different (*p* ≤ 0.05).

## Data Availability

The data presented in this study are available on request from the corresponding author.

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
