# Peer review of "Preferences in ‘Jalapeño’ Pepper Attributes: A Choice Study in Mexico"

_foods, 2021, doi:10.3390/foods10123111_

Round 1
Reviewer 1 Report
The paper is clear and interesting. I have only some minor comments:
- the first row of the abstract should be modified, presenting the subjects (i.e. the product)
- line 54: get rid of "to"
- line 67: get rid of "to pay" (it appears twice)
- line 128: please, check when you mention "the term n σ..." (specify which you mention)
- line 157-159: please, better clarify LMA and how you derive 46.7. According to the theory, I would expect 36 combinations in a full factorial design.
- table 5: please, use the notes to specify everything that is included in the table, in order to make it more readable
- line 199: show instead of shown
- paragraph from line 199 to line 209: please, report WTP for size and pungency adequately
- line 225: please, check the percentage and report them coherently with what reported in table 6
- from line 241 to line 280: for a better readiness, I suggest to report (in brackets) the name of the variables that you explain
Author Response
The paper is clear and interesting. I have only some minor comments:
- The first row of the abstract should be modified, presenting the subjects (i.e. the product).
R: “of ‘Jalapeño’ peppers” was added (page 1, line 12).
- Line 54: get rid of "to"
R: “to” was deleted (page 2; line 59).
- Line 67: get rid of "to pay" (it appears twice)
R: The second “to take” was deleted (page 2; line 72).
- Line 128: please, check when you mention "the term n σ..." (Specify which you mention)
- The term was revised (page 4; line 135).
- Line 157-159: please, better clarify LMA and how you derive 46.7. According to the theory, I would expect 36 combinations in a full factorial design.
R: According to the Theory of the Discrete choice experiment a full factorial design with all combination will result in LA combination where A is the attributes and L are the number of levels resulting in (32×41=36) as we have two attributes with three levels and one attribute with four levels as the reviewer highlight. Thus, taking into account the number of alternative M in each choice set (which is 3), the total number of possible combinations in this case is LMA (363 = 46,656 alternative combinations) (page 5; line 165).
- Table 5: please, use the notes to specify everything that is included in the table, in order to make it more readable
R: See explanations at the end of table 5 (page 8; lines 207-209).
- Line 199: show instead of shown
R: This was done as suggested (page 8; line 211).
- Paragraph from line 199 to line 209: please, report WTP for size and pungency adequately
- See right values on page 8, lines 213 and 218.
- Line 225: please, check the percentage and report them coherently with what reported in table 6.
R: See the percentages on page 8, lines 237 and 238.
- From line 241 to line 280: for a better readiness, I suggest to report (in brackets) the name of the variables that you explain
R: We clarify this misunderstanding in table 7, therefore segment was replaced for “variable” (page 9).

Reviewer 2 Report
Please find attached the review file.

Author Response
The paper entitled “Preferences in ‘Jalapeño’ Pepper Attributes: A Choice Study in Mexico”
identifies ‘Jalapeño’ quality attributes to determine consumer preferences and willingness to pay, based on socioeconomic characteristic. The authors carried out a nationwide face-to-face survey and used the discrete choice experiment method. The findings reveal heterogeneity among ‘Jalapeño’ consumers in Mexico with three consumer profiles with respect to price: price sensitivity, non-demanding (indifferent), and selective. These findings allow growers to focus efforts into less demanding segments or develop new market strategies.
- Abstract: the authors recommend “farmers must grow the correct varieties with appropriate agronomic management to achieve these results”. This recommendation is not based on the paper results since no variety differences or agronomic management is considered in the analysis.
R: We have modified the idea according with your comment (page 1; line 22-24).
- Introduction
Line 32: “This increase is consistent with the goals of world food security programs [2]” it is not clear how the Worldwide increase of chili yield is consistent with food security programs; can you add more explanation.
R: According with your comment, we have completed the idea (page 1; lines 34-35).
- Line 44: “In addition, new challenges have emerged as supply chains shift their focus to satisfy
consumer demands directly”. Which challenges want the authors to mention here, can you
specify?
R: We added information regarding the current pandemic (page 2; lines 47-49).
- Line 46: “Supply and added-value strategies require more information on consumer behavior [5]”. To which additional information on consumer behavior are the authors referring, can you specify?
We have completed the idea with the previous (page 2; line 50-51).
- 2.2 The Discrete Choice Experiment: Theory and modelling approach
Why did you use discrete choice experiment method? What are advantages and disadvantages of other methods (e.g. AHP)?
R: We used the DCE rather than AHP because our objective was to estimate the parameter “willing to pay” directly from the coefficients. The advantages associated with DCE are: a) better fits the data, b) reduce the incidence of excessively large estimated values and, c) provides the analyst with greater control over the distribution of the WTP
- 2. Materials and Methods
2.1. Data collection and sampling
There is missing information regarding the procedure that can be added to 2.1. Data collection and sampling:
- Which sampling method did you use and how you select individuals that participate in the choice (page 86-88)? Randomly?
R: The sampling method was randomly (page 2; line 90) and the participants considered for this study were adults over 20-year (page 2, line 88).
- How you contact selected individuals (1,200 select individuals) in order to run the questionnaires with the choice explanation?
R: This was done randomly (see page 2; line 90)
Did researchers read the questions to the participants? In the abstract you speak about face-to-face survey, where was the survey conducted?
R: We added this on page 3, lines 98 -101.
How you manage the step from the selection based on national data to the execution (face-to-face)?
R: This was done randomly (see page 2; line 90)
How many consumers were recruited for the study at the beginning and how many completed the questionnaire, please clarify.
R: See explantions on page 2, lines 89-90.
On average how long did the survey take participants to complete?
R: Please read page 3, line 100-101.
- 2.4. The Discrete Choice experiment: Empirical applications
In Table 2. Attributes and levels from ‘Jalapeño’ fruit. You consider 4 price levels. However there is just 5cents USD/kg difference between the first level and the second of the price (L1.1= 51 cents USD/kg ; L1.2= 56 cents) while the difference to the other prices levels L1.3 and L1.4 are more distinguished (L1.3 =1.07 USD/kg L1.4=2.05 USD/kg), how you justify this choice and how the consumers could distinguish between the first two levels of prices in a choice set where there is more variation in other attributes.
- The Jalapeño prices were obtained from different markets on wheels, markets, and supermarkets, therefore the values given in table 2 are the average from the lowest to the highest prices found commercially.
- 2.3. Latent class analysis
Why you decide to use Latent class to analyze Heterogeneity in consumption behavior comparing to other methods. Add the justification to 2.3.
R: The explanation is given on page 4, lines 146-148.
- 3.4. Profile of consumer segments.
How much percentage represent each segment on the total sample? Please introduce this
information in Table 7. Key variables describing the consumers for each segment.
R: The information is given in table 7 (page 9)
- You don’t discuss any limitations of your study and the used method. I would highly recommend to do so.
R: The suggestion regarding limitations forms our experiment is given on page 10, lines 296-298.
Regarding method, we don’t have any comments because this statistical tool was used to reach our objectives compared to AHP.
